# Design and Synthesis of Some New Furan-Based Derivatives and Evaluation of In Vitro Cytotoxic Activity

**DOI:** 10.3390/molecules27082606

**Published:** 2022-04-18

**Authors:** Syed Nasir Abbas Bukhari, Hasan Ejaz, Mervat A. Elsherif, Kashaf Junaid, Islam Zaki, Reham E. Masoud

**Affiliations:** 1Department of Pharmaceutical Chemistry, College of Pharmacy, Jouf University, Sakaka 72388, Saudi Arabia; 2Department of Clinical Laboratory Sciences, College of Applied Medical Sciences, Jouf University, Sakaka 72388, Saudi Arabia; hetariq@ju.edu.sa (H.E.); kjunaid@ju.edu.sa (K.J.); 3Chemistry Department, College of Science, Jouf University, Sakaka 72388, Saudi Arabia; maelsherif@ju.edu.sa; 4Pharmaceutical Organic Chemistry Department, Faculty of Pharmacy, Port Said University, Port Said 42526, Egypt; 5Clinical Pharmacology Department, Faculty of Medicine, Port Said University, Port Said 42526, Egypt; rehame_ph@yahoo.com

**Keywords:** cytotoxicity, MCF-7, furan, apoptosis, cell cycle analysis, Bax, Bcl2

## Abstract

New furan-based derivatives have been, designed, synthesized, and evaluated for their cytotoxic and tubulin polymerization inhibitory activities. DNA flow cytometric study of pyridine carbohydrazide **4** and *N*-phenyl triazinone **7** demonstrated G_2_/M phase cell cycle disruptions. Accumulation of cells in the pre-G1 phase and positive annexin V/PI staining, which may be caused by degeneration or fragmentation of the genetic components, suggested that cell death occurs via an apoptotic cascade. Furthermore, compounds **4** and **7** had a strong pro-apoptotic impact through inducing the intrinsic mitochondrial mechanism of apoptosis. This mechanistic route was verified by an ELISA experiment that indicated a considerable rise in the levels of p53 and Bax and a drop in the level of Bcl-2 when compared with the control.

## 1. Introduction

In cancer, cells proliferate and spread uncontrollably due to inherent and acquired genetic abnormalities that affect cell proliferation and survival [1,2,3]. Breast cancer is the most prevalent kind of carcinoma in women and the second leading cause of cancer-related mortality among them [4,5,6]. Numerous investigations have indicated that breast cancer cells have an abnormally high mitotic rate as a result of increased microtubule synthesis [7,8]. Thus, inhibiting tubulin polymerization in MCF-7 cancer cells may provide a significant target for developing new anticancer drugs [9,10,11].

Microtubules serve as a valuable and strategic molecular target for the development of anticancer medicines that bind to specific locations in tubulin subunits [12,13,14]. Since inhibition of assembly or disassembly of tubulin/microtubule system leads to similar cellular events, drugs that target either process have been developed [15]. Noscapine (Figure 1A) and CC-5079 (Figure 1B) possessing dimethoxyphenyl (DMP) moiety were identified as potent antitumor molecules and a cause of apoptosis in many cell types [16,17]. Indibulin (Figure 1C) is an orally active antimitotic drug that is effective against a variety of human cell lines, including taxane-resistant cells [18]. Additionally, it was discovered that dienone molecule (Figure 1D) stabilizes tubulin similarly to docetaxel [19].

Furan nucleus is a pharmacologically active pharmacophoric entity and compounds containing furan ring have received increasing attention owing to their promising anticancer activity [20,21,22,23,24,25]. Furan-2-carboxamide molecule (Figure 1E), for example, showed powerful antiproliferative activity against a panel of cancer cell lines in vitro [26]. Compound (Figure 1F) with furan group exhibited potent cytotoxic activities at the nanomolar level against various human cancer cell lines (IC_50_ = 2.9 nM, NCI-H460) [27]. In addition, furan derivative (Figure 1G) produced a significant reduction in cellular microtubule of MCF-7 cells with excellent β-tubulin polymerization inhibition activity (Figure 1) [28].

Prompted by the above-mentioned studies and in our continuous program in the search for new candidates with anticancer agents [29,30,31,32], in this study, we describe the synthesis and biological assessment of a novel class of furan-based derivatives that may have enhanced anticancer activity (Figure 2). The newly synthesized compounds were tested against MCF-7 breast cancer and MCF-10A normal breast cell lines for their antiproliferative properties. In addition, the most active molecules were further evaluated for the inhibitory activity against tubulin as a potential target for molecular mechanism. Furthermore, the most active compounds were evaluated for cell cycle analysis and apoptosis induction.

## 2. Results and Discussion

### 2.1. Chemistry

Figure 1 depicts the synthesis of furan-based compounds **2**–**7**. According to a previously published methodology [33], the main starting material 2-(3,4-dimethoxyphenyl)-4-(furan-2-ylmethylene) oxazol-5-one (**1**) was synthesized using 3,4-dimethoxyhippuric acid and furan in the presence of acetic anhydride containing anhydrous pyridine. IR spectrum of compound **1** revealed the presence of strong absorption band at 1775 cm^−1^ characteristic of carbonyl group (C=O) of lactone ring. Oxazolone **1** and methanol were reacting in the presence of triethyl amine (Et_3_N) to produce the next methyl ester derivative **2**. The ^1^H-NMR spectrum of **2** revealed an exchangeable proton (NH) at δ 9.83 ppm, in addition to new signal at δ 3.72 ppm related to the methyl group of methyl ester (COOCH_3_) function. The oxazolone 1 and primary aromatic amine reaction in ethanol at reflux temperature was used to synthesize the acrylamide molecules **3a**,**b**. ^1^H-NMR and ^13^C-NMR spectrum data corroborated the structures of the synthesized compounds **3a** and **b**. The ^1^H-NMR spectra displayed the presence of two exchangeable protons (2NH) at δ 9.82–9.96 ppm and the presence of new proton signals assigned to the tolyl group. In addition, ^13^C-NMR spectra showed additional aromatic signals attributed to tolyl carbons. Moreover, synthesis of 4-pyridine carbohydrazide derivative **4** was accomplished by the reaction of **1** with 4-pyridine carbohydrazide. Three exchangeable signals at δ 9.80, 10.31, and 10.79 ppm corresponding to three NH protons were found in the ^1^H-NMR spectrum; two additional aromatic signals with four protons total attributed to the pyridine group were also found. ^13^C-NMR spectrum of compound **4** confirmed the carbon skeleton due the presence of new carbon signals assigned to carbonyl group of the hydrazide and pyridine groups. In this investigation, the imidazole derivatives **5a** and **5b** were synthesized by heating a combination of equimolar quantities of 1 with the appropriate primary aromatic amines in glacial acetic acid in the presence of anhydrous sodium acetate (NaOAc) as reported previously [34]. The aromatic ring protons of compounds **5a**,**b** displayed a characteristic set of proton signals corresponding to tolyl group protons. Compounds **5a**,**b** exhibited the existence of extra carbon signals associated with tolyl carbons in their ^13^C-NMR spectra. Upon applying the same previous procedure using thiosemicarbazide instead of primary aromatic amines, the target compound **6** was obtained. The structure of new compound **6** was confirmed based on spectral data ^1^H-NMR and ^13^C-NMR spectra. ^1^H-NMR spectrum of compound **6** revealed the presence of new proton signals related to NH_2_ and NH groups. ^13^C-NMR spectrum of compound **6** showed an additional carbon signal at δ 182.95 ppm attributed to thiocarbonyl carbon (C=S). On the other hand, compound **7** was prepared by heating at reflux the key intermediate **1** with phenyl hydrazine in ethanol at reflux temperature. ^1^H-NMR spectrum of the titled compound **7** revealed the presence of new signal at δ 9.06 ppm indicating the presence of NH proton of triazinone ring as well as set of signals appearing between δ 6.66–8.01 ppm that corresponded to phenyl protons. Compound **7**’s ^13^C-NMR spectra revealed the existence of extra carbon signals attributable to phenyl carbons.

### 2.2. Biological Evaluation and Mechanistic Studies

#### 2.2.1. Cytotoxic Activity against Breast Cancer Cell Lines (MCF-7)

Compounds **2**–**7** were tested for cytotoxicity against MCF-7 breast cancer cells in a conventional MTT assay. From the mean values of three different triplicates, the percentage of relative viability and the half maximum inhibitory concentration IC_50_ were computed. This study’s reference control was staurosporin (STU). The treatment of breast cancer MCF-7 cells with gradual concentrations of furan-based compounds **2**–**7** indicates that two compounds—**4** and **7**—exhibited good cytotoxic activity with IC_50_ values 4.06 and 2.96 μM, respectively, compared with STU (Table 1). Compounds **4** and **7** were, on the other hand, evaluated for their cytotoxic activity against a normal breast cell line (MCF-10A). The results show that the tested molecules exhibited higher IC_50_ values against normal breast cell line. Furthermore, the selectivity index (SI) of MCF-7 cells for compounds **4** and **7** using IC_50_ values was calculated. The SI of compounds **4** and **7** was 7.33 and 7.47 times, respectively, higher than that for MCF-7 cells. Therefore, they can be considered selective cytotoxic molecules against MCF-7 cancer cell line and safe toward the normal breast cells.

#### 2.2.2. In Vitro β-Tubulin Polymerization Assay

Compounds **4** and **7** were tested in vitro for their ability to inhibit the polymerization of β-tubulin. From the obtained results, compounds **4** and **7** caused 53% and 71% inhibition, respectively, at a concentration equal to their IC_50_ concentration compared with untreated control (Figure 3). According to these results, Compounds **4** and **7** are involved in tubulin polymerization inhibition activity.

#### 2.2.3. Cell Cycle Analysis

Cell cycle analysis was used to elucidate the mechanism of cell proliferation inhibition action [35]. As a result, compounds **4** and **7** at their IC_50_ concentration (µM) were treated with breast MCF-7 cells for 48 h before the cells were examined using flow cytometry. Cell cycle distribution was interrupted in this cell line by compounds **4** and **7** based on the results of this study. Cells at pre-G1 increased by 3.64- and 5.86-fold, respectively, after treatment with compounds **4** and **7**. In addition, compounds **4** and **7** significantly increased the number of cells in the G_2_/M phase by 2.42- and 4.48-fold, respectively, compared with no control treatment. The results from cell cycle profile analysis revealed that compounds **4** and **7** resulted in apoptosis and G2/M arrest of MCF-7 cells (Figure 4).

#### 2.2.4. Annexin V-FITC/PI and Detection of Apoptosis

It is possible to determine the number of dead cells in a sample using the annexin V/PI dual staining experiment, which may be performed using flow cytometry [36]. To quantify the percentage (%) of apoptosis induced by compounds **4** and **7** in breast MCF-7 cells, annexin V/PI dual staining was carried out, and the results are shown in Figure 5. From the obtained results, the treated cells displayed an increased percentage of Annexin V dye in the early apoptotic stage by 2.18- and 3.33-folds, respectively, more than the untreated control. Furthermore, the annexin-V positive cells in the late apoptotic stage rose by 9.31- and 16.66-fold in the treated cells compared with the untreated reference. These results show that compounds **4** and **7** can be considered as apoptosis inducers.

#### 2.2.5. Effect of Compounds **4** and **7** on the Level of p53/Bax/Bcl-2

The apoptotic markers p53, Bax, and Bcl-2 were found in the MCF-7 cells treated with compounds **4** and **7** for 48 h at their IC_50_ concentrations (µM). The three well-known tumor suppressor genes—p53, Bax, and Bcl-2—play a vital part in the apoptotic processes of the cells. Cellular apoptosis can be induced by overexpression of these proteins. Compounds **4** and **7** increased the amount of p53 in MCF-7 cells by 6.95- and 12.02-fold, respectively, as compared with untreated control cells. Compounds **4** and **7** had Bax levels that were 3.15 and 3.89 times higher than the untreated control, respectively. While this was going on, the Bcl-2 concentration was 2.38 and 4.36 times lower than in the untreated control group. Apoptosis may play a role in compounds **4** and **7**-induced cancer cell death and cytotoxicity, according to these findings (Figure 6).

#### 2.2.6. Molecular Docking Study

To better understand how tubulin polymerization inhibitory activity is mediated, compounds **4** and **7** were docked into the crystal structure of tubulin (PDB entry: 1SA0). Hydrazide C=O group and pyridine moiety of synthetic derivative **4** are shown to interact with amino acids Tyr 228 and Asn 228 through hydrogen bonding in Figure 7. The hydrophobic side chain of Tyr 224 interacts hydrophobically with its pyridine group. Furthermore, the studies showed a docking energy of 21.72 kcal/mol. Compound **7** on the other hand interacts with Glu 183 via the NH moiety of the triazine as an H. bond donor. Gly 144 interacts with its methoxy group because it is a hydrogen bond acceptor. Compound **7** also has a furan moiety, which has hydrophobic interactions with amino acid Tyr 224. Compound **7**’s hydrophobicity led to a docking score of −26.33 kcal/mol.

## 3. Experimental

### 3.1. Chemistry

#### 3.1.1. General

Electrothermal digital melting point equipment was used to measure the melting points of compounds in open capillaries. We used a Bruker 400 MHz NMR spectrometer to capture the ^1^H and ^13^C NMR spectra, and the peak positions are provided in ppm downfield of the internal standard, tetramethylsilane (TMS). An Elementar, Vario El, Microanalytical unit, Cairo, Egypt, was used to perform elemental analyses, and the results were determined to be within 0.4 percent of the theoretical values. Commercially available chemicals and reagents were procured and employed right away.

#### 3.1.2. General Procedure for the Synthesis of Methyl 2-(3,4-dimethoxybenzamido)-3-(furan-2-yl)acrylate (**2**)

For 1 h, a mixture of compound 1 (0.299 g, 1 mmol) and Et_3_N (0.14 mL, 1 mmol) was refluxed in methanol (20 mL). The reaction mixture was poured onto crushed ice/water and allowed to remain at room temperature for 2 h after cooling. The obtained solid was filtered and crystallized from methanol to obtain compound **2**.

White powder (0.229 g, 69.20%), m.p. 135–137 °C. ^1^H-NMR (400 MHz, DMSO-*d_6_*): δ 3.72 (s, 3H, OCH_3_), 3.83 (s, 3H, OCH_3_), 3.84 (s, 3H, OCH_3_), 6.63 (dd, *J* = 3.4, 1.8 Hz, 1H, furan CH), 6.85 (d, *J* = 3.4 Hz, 1H, arom. CH), 7.24 (s, 1H, olefinic CH), 7.08 (d, *J* = 8.5 Hz, 1H, arom. CH), 7.58 (d, *J* = 1.8 Hz, 1H, furan CH), 7.66 (dd, *J* = 8.4, 1.9 Hz, 1H, arom. CH), 7.84 (d, *J* = 1.5 Hz, 1H, furan CH), 9.83 (s, 1H, NH, D_2_O exchange) ppm (Appendix A). ^13^C-NMR (101 MHz, DMSO): δ 52.67 (OCH_3_), 56.06 (OCH_3_), 56.15 (OCH_3_), 111.40 (C3 methyl acrylate), 111.50 (C5 dimethoxy benzamide), 113.02 (C3 furan), 116.25 (C4 furan), 120.73 (C2 dimethoxy benzamide), 121.67 (C6 dimethoxy benzamide), 124.16 (C1 dimethoxy benzamide), 126.01 (C2 methyl acrylate), 145.96 (C5 furan), 148.83 (C2 furan), 149.69 (C3 dimethoxy benzamide), 152.29 (C4 dimethoxy benzamide), 165.53 (C1 methyl acrylate), 165.65 (C=O dimethoxy benzamide) ppm. Anal. Calcd. for C_17_H_17_NO_6_ (331.23): C, 61.63; H, 5.17; N, 4.23. Found: C, 61.77; H, 5.02; N, 4.12 (Appendix A).

#### 3.1.3. General Procedure for the Synthesis of N-(1-(furan-2-yl)-3-oxo-3-(arylamino)prop-1-en-2-yl)-3,4-dimethoxybenzamides **3a**,**b**

A suitable primary aromatic amine (1 mmol) was added to a suspension of oxazolone 1 (0.299 g, 1 mmol) in pure ethanol (20 mL). For four hours, the reaction mixture was re-fluxed. Following cooling of the reaction mixture, the crystals formed were recovered and recrystallized from 100% ethanol to provide the pure compound **3a**,**b**.

##### N-(1-(furan-2-yl)-3-oxo-3-(m-tolylamino)prop-1-en-2-yl)-3,4-dimethoxybenzamide (**3a**)

White powder (0.266 g, 65.58%), m.p. 181–183 °C. ^1^H-NMR (400 MHz, DMSO-*d*_6_) δ 2.28 (s, 3H, CH_3_), 3.84 (s, 3H, OCH_3_), 3.85 (s, 3H, OCH_3_), 6.60 (dd, *J* = 3.4, 1.8 Hz, 1H, furan CH), 6.74 (d, *J* = 3.4 Hz, 1H, arom. CH), 6.89 (d, *J* = 7.5 Hz, 1H, arom. CH), 7.06 (s, 1H, olefinic CH), 7.10 (d, *J* = 8.5 Hz, 1H, arom. CH), 7.19 (t, *J* = 7.8 Hz, 1H, arom. CH), 7.48–7.57 (m, 2H, arom. CH), 7.64 (d, *J* = 1.9 Hz, 1H, furan CH), 7.70 (dd, *J* = 8.4, 1.9 Hz, 1H, arom. CH), 7.79 (d, *J* = 1.4 Hz, 1H, furan CH), 9.82 (s, 1H, NH, D_2_O exchange), 9.96 (s, 1H, NH, D_2_O exchange) ppm (Appendix A). ^13^C-NMR (101 MHz, DMSO): δ 21.65 (CH_3_), 56.08 (OCH_3_), 56.15 (OCH_3_), 111.42 (C3 acrylamide), 111.69 (C5 dimethoxy benzamide), 112.78 (C3 furan), 114.23 (C4 furan), 116.79 (C2 dimethoxy benzamide), 117.79 (C6 tolyl), 121.12 (C6 dimethoxy benzamide), 121.87 (C4 tolyl), 124.56 (C2 tolyl), 126.34 (C1 dimethoxy benzamide), 128.78 (C5 tolyl), 128.93 (C2 acrylamide), 138.06 (C3 tolyl), 139.57 (C1 tolyl), 144.90 (C5 furan), 148.73 (C2 furan), 150.26 (C3 dimethoxy benzamide), 152.21 (C4 dimethoxy benzamide), 164.04 (C=O dimethoxy benzamide), 165.62 (C=O acrylamide). Anal. Calcd. for C_23_H_22_N_2_O_5_ (406.43): C, 67.97; H, 5.46; N, 6.89. Found: C, 68.12; H, 5.58; N, 6.77 (Appendix A).

##### N-(1-(furan-2-yl)-3-oxo-3-(p-tolylamino)prop-1-en-2-yl)-3,4-dimethoxybenzamide (**3b**)

White powder (0.260 g, 63.92%), m.p. 187-189 °C. ^1^H-NMR (400 MHz, DMSO-*d*_6_): δ 2.26 (s, 3H, CH_3_), 3.84 (s, 3H, OCH_3_), 3.85 (s, 3H, OCH_3_), 6.60 (dd, *J* = 3.5, 1.8 Hz, 1H, furan CH), 6.73 (d, *J* = 3.4 Hz, 1H, arom. CH), 7.04–7.16 (m, 4H, arom. CH and olefinic CH), 7.59 (d, *J* = 8.4 Hz, 2H, arom. CH), 7.64 (d, *J* = 2.1 Hz, 1H, furan CH), 7.70 (dd, *J* = 8.4, 2.1 Hz, 1H, arom. CH), 7.79 (d, *J* = 1.8 Hz, 1H, furan CH), 9.83 (s, 1H, NH, D_2_O exchange), 9.96 (s, 1H, NH, D_2_O exchange) (Appendix A). ^13^C-NMR (101 MHz, DMSO): δ 20.47 (CH_3_), 55.59 (OCH_3_), 55.66 (OCH_3_), 110.93 (C3 acrylamide), 111.21 (C5 dimethoxy benzamide), 112.28 (C3 furan), 113.70 (C4 furan), 116.38 (C2 dimethoxy benzamide), 120.15 (C2,6 tolyl), 121.38 (C6 dimethoxy benzamide), 125.88 (C3,5 tolyl), 128.45 (C1 dimethoxy benzamide), 128.84 (C2 acrylamide), 132.30 (C1 tolyl), 136.65 (C4 tolyl), 144.39 (C5 furan), 148.24 (C2 furan), 149.79 (C3 dimethoxy benzamide), 151.71 (C4 dimethoxy benzamide), 163.42 (C=O dimethoxy benzamide), 165.14 (C=O acrylamide). Anal. Calcd. for C_23_H_22_N_2_O_5_ (406.43): C, 67.97; H, 5.46; N, 6.89. Found: C, 67.82; H, 5.63; N, 7.04 (Appendix A).

#### 3.1.4. General Procedure for the Synthesis of N-(1-(furan-2-yl)-3-(2-isonicotinoylhydrazinyl)-3-oxoprop-1-en-2-yl)-3,4-dimethoxybenzamide (**4**)

4-Pyridine carbohydrazide (0.137 g, 1 mmol) was added to a suspension of compound 1 (0.299 g, 1 mmol) in 100% ethanol (20 mL). For four hours, the reaction mixture was refluxed and then concentrated at decreased pressure. The obtained solid product was filtered and crystallized from 100% ethanol after cooling to obtain the named compound **4**.

White powder (0.203 g, 46.51%), m.p. 232–234 °C. ^1^H-NMR (400 MHz, DMSO-*d*_6_) δ 3.84 (s, 6H, 2OCH_3_), 6.60 (s, 1H, furan CH), 6.69–6.84 (m, 1H, arom. CH), 7.09 (d, *J* = 8.3 Hz, 1H, arom. CH), 7.22 (s, 1H, olefinic CH), 7.60–7.76 (m, 2H, furan CH), 7.74–7.88 (m, 3H, arom. CH), 8.76 (d, *J* = 4.6 Hz, 2H, arom. CH), 9.80 (s, 1H, NH, D_2_O exchange), 10.31 (s, 1H, NH, D_2_O exchange), 10.79 (s, 1H, NH, D_2_O exchange) ppm (Appendix A). ^13^C-NMR (101 MHz, DMSO): δ 56.09 (OCH_3_), 56.14 (OCH_3_), 111.36 (C3 acrylamide), 111.82 (C5 dimethoxy benzamide), 112.85 (C3 furan), 114.77 (C4 furan), 118.71 (C2 dimethoxy benzamide), 121.82 (C2,6 pyridine), 121.93 (C6 dimethoxy benzamide), 126.52 (C1 dimethoxy benzamide), 126.73 (C2 acrylamide), 140.09 (C1 pyridine), 145.20 (C5 furan), 148.63 (C2 furan), 150.04 (C3 dimethoxy benzamide), 150.86 (C3,5 pyridine), 152.14 (C4 dimethoxy benzamide), 164.49 (C=O hydrazide), 164.58 (C=O), 165.86 (C=O dimethoxy benzamide). Anal. Calcd. for C_22_H_20_N_4_O_6_ (436.42): C, 60.55; H, 4.62; N, 12.84. Found: C, 60.72; H, 4.71; N, 12.66 (Appendix A).

#### 3.1.5. General Procedure for the Synthesis of 1-Aryl-2-(3,4-dimethoxyphenyl)-4-(furan-2-ylmethylene)-1H-imidazol-5(4H)-ones **5a**,**b**

Oxazolone 1 (0.299 g, 1 mmol), primary aromatic amines (1 mmol), and anhydrous sodium acetate (0.098 g, 1.2 mmol) were refluxed for 10–12 h in glacial acetic acid (20 mL). Black powder was separated from 100% ethanol by pouring it on crushed ice/water, then filtered, rinsed with water, and crystallized.

##### 2-(3,4-Dimethoxyphenyl)-4-(furan-2-ylmethylene)-1-m-tolyl-1H-imidazol-5(4H)-one (**5a**)

Black powder (0.225 g, 58.09%), m.p. 143–145 °C. ^1^H-NMR (400 MHz, Chloroform-*d*) δ 2.36 (s, 3H, CH_3_), 3.65 (s, 3H, OCH_3_), 3.88 (s, 3H, OCH_3_), 6.62 (s, 1H, furan CH), 6.77 (d, *J* = 8.5 Hz, 1H, arom. CH), 6.98 (d, *J* = 7.6 Hz, 1H, arom. CH), 7.08 (d, *J* = 17.2 Hz, 2H, arom. CH), 7.14–7.24 (m, 3H, olefinic and arom. CH), 7.29–7.34 (m, 1H, furan CH), 7.49 (d, *J* = 3.3 Hz, 1H, furan CH), 7.65 (s, 1H, arom. CH) ppm (Appendix A). ^13^C-NMR (101 MHz, Chloroform-*d*): δ 21.38 (CH_3_), 55.68 (OCH_3_), 56.01 (OCH_3_), 110.61 (C3 furan), 111.91 (C2 dimethoxyphenyl), 113.50 (C5 dimethoxyphenyl), 115.07 (C4 furan), 117.11 (C1 dimethoxyphenyl), 118.96 (C6 dimethoxyphenyl), 121.08 (C4 tolyl), 123.31 (C6 tolyl), 124.87 (C5 tolyl), 128.29 (C2 tolyl), 129.37 (C olefinic), 135.13 (C1 tolyl), 135.98 (C4 imidazole), 139.77 (C3 tolyl), 145.99 (C5 furan), 148.45 (C3 dimethoxyphenyl), 151.43 (C4 dimethoxyphenyl), 151.82 (C2 furan), 159.51 (C2 imidazole), 170.28 (C=O imidazole) ppm. Anal. Calcd. for C_23_H_20_N_2_O_4_ (388.42): C, 71.12; H, 5.19; N, 7.21. Found: C, 71.24; H, 4.93; N, 7.04 (Appendix A).

##### 2-(3,4-Dimethoxyphenyl)-4-(furan-2-ylmethylene)-1-p-tolyl-1H-imidazol-5(4H)-one (**5b**)

Black powder (0.238 g, 61.22%), m.p. 158–160 °C. ^1^H-NMR (400 MHz, Chloroform-*d*) δ 2.46 (s, 3H, CH_3_), 3.74 (s, 3H, OCH_3_), 3.95 (s, 3H, OCH_3_), 6.69 (s, 1H, furan CH), 6.84 (d, *J* = 8.5 Hz, 1H, arom. CH), 7.11–7.23 (m, 3H, olefinic and arom. CH), 7.26–7.29 (m, 1H, arom. CH), 7.29–7.39 (m, 3H, arom. CH), 7.56 (d, *J* = 3.2 Hz, 1H, furan CH), 7.71 (d, *J* = 11.8 Hz, 1H, furan CH) ppm (Appendix A). ^13^C-NMR (101 MHz, Chloroform-*d*): δ 21.20 (CH_3_), 55.61 (OCH_3_), 55.94 (OCH_3_), 110.52 (C3 furan), 111.87 (C2 dimethoxyphenyl), 113.42 (C5 dimethoxyphenyl), 115.00 (C4 furan), 118.85 (C1 dimethoxyphenyl), 121.09 (C6 dimethoxyphenyl), 123.21 (C olefinic), 127.44 (C2,6 tolyl), 130.14 (C3,5 tolyl), 132.49 (C1 tolyl), 135.98 (C4 imidazole), 138.52 (C4 tolyl), 145.90 (C5 furan), 148.42 (C3 dimethoxyphenyl), 151.38 (C4 dimethoxyphenyl), 151.73 (C2 furan), 159.58 (C2 imidazole), 170.29 (C=O imidazole) ppm. Anal. Calcd. for C_23_H_20_N_2_O_4_ (388.42): C, 71.12; H, 5.19; N, 7.21. Found: C, 70.90; H, 5.19; N, 7.07 (Appendix A).

#### 3.1.6. General Procedure for the Synthesis of 3-(3,4-Dimethoxyphenyl)-5-(furan-2-ylmethylene)-6-oxo-5,6-dihydro-1,2,4-triazine-2(1H)-carbothioamide (**6**)

A mixture of oxazolone 1 (0.299 g, 1 mmol), thiosemicarbazide (0.091 g, 1 mmol), and anhydrous sodium acetate (0.098, 1.2 mmol) in glacial acetic acid (20 mL) was refluxed for 8 h. Crushed ice and water was added to the reaction mix when it reached the TLC completion point. In order to obtain the pure chemical **6**, the crude methanol product was filtered, dried, and crystallized.

Red powder (0.213 g, 57.35%), m.p. 202–204 °C. ^1^H-NMR (400 MHz, Chloroform-*d*) δ 3.89 (s, 3H, OCH_3_), 3.98 (s, 3H, OCH_3_), 6.05–6.46 (m, 2H, NH_2_), 6.64 (s, 1H, furan CH), 6.97 (d, *J* = 8.7 Hz, 1H, arom. CH), 7.11 (s, 1H, olefinic CH), 7.49 (s, 1H, furan CH), 7.65 (d, *J* = 19.1 Hz, 2H, arom. CH), 7.79 (d, *J* = 8.7 Hz, 1H, furan CH), 8.84 (s, 1H, NH) ppm (Appendix A). ^13^C-NMR (101 MHz, Chloroform-*d*): δ 56.14 (OCH_3_), 56.17 (OCH_3_), 110.20 (C3 furan), 111.01 (C2 dimethoxyphenyl), 113.65 (C5 dimethoxyphenyl), 117.05 (C4 furan), 117.92 (C1 dimethoxyphenyl), 119.58 (C6 dimethoxyphenyl), 122.85 (C olefinic), 130.70 (C5 triazine), 146.41 (C5 furan), 149.26 (C3 dimethoxyphenyl), 150.61 (C4 dimethoxyphenyl), 153.03 (C2 furan), 162.93 (C3 triazine), 167.35 (C=O triazine), 182.95 (C=S) ppm. Anal. Calcd. for C_17_H_16_N_4_O_4_S (372.40): C, 54.83; H, 4.33; N, 15.04. Found: C, 54.97; H, 4.52; N, 14.86 (Appendix A).

#### 3.1.7. General Procedure for the Synthesis of 3-(3,4-Dimethoxyphenyl)-5-(furan-2-ylmethylene)-2-phenyl-2,5-dihydro-1,2,4-triazin-6(1H)-one (**7**)

Phenyl hydrazine (0.098 mL, 1 mmol) was added to a solution of compound 1 (0.299 g, 1 mmol) in absolute ethanol (20 mL). Following a 6 h reflux process, the reaction mixture was concentrated at decreased pressure. It was then filtered and crystallized from DMF/H2O (1:1) in order to yield the pure compound **7** after cooling.

Black powder (0.291 g, 74.82%), m.p. 177–179 °C. ^1^H-NMR (400 MHz, DMSO-*d*_6_) δ 3.69 (s, 3H, OCH_3_), 3.81 (s, 3H, OCH_3_), 6.66–6.76 (m, 2H, furan CH and arom. CH), 6.82 (tt, *J* = 5.2, 2.2 Hz, 2H, arom. CH), 7.04 (s, 1H, olefinic CH), 7.07 (d, *J* = 8.7 Hz, 1H, arom. CH), 7.17–7.27 (m, 2H, arom. CH), 7.63 (d, *J* = 3.5 Hz, 1H, arom. CH), 7.76 (t, *J* = 1.6 Hz, 1H, furan CH), 7.88 (dd, *J* = 8.7, 1.9 Hz, 1H, arom. CH), 8.01 (d, *J* = 1.7 Hz, 1H, furan CH), 9.06 (s, 1H, NH, D_2_O exchange) ppm (Appendix A). ^13^C-NMR (101 MHz, DMSO-*d*_6_): 55.30 (OCH_3_), 55.67 (OCH_3_), 111.37 (C3 furan), 111.51 (C2 dimethoxyphenyl), 112.08 (C2,6 phenyl), 113.62 (C5 dimethoxyphenyl), 114.01 (C4 furan), 119.28 (C1 dimethoxyphenyl), 119.63 (C4 phenyl), 119.98 (C6 dimethoxyphenyl), 122.56 (C olefinic), 129.32 (C3,5 phenyl), 133.61 (C1 phenyl), 146.45 (C5 triazine), 146.90 (C5 furan), 148.34 (C3 dimethoxyphenyl), 150.50 (C4 dimethoxyphenyl), 152.35 (C2 furan), 159.24 (C3 triazine), 168.78 (C=O triazine). Anal. Calcd. for C_22_H_19_N_3_O_4_ (389.41): C, 67.86; H, 4.92; N, 10.79. Found: C, 68.04; H, 5.08; N, 10.63 (Appendix A).

### 3.2. Pharmacological Studies

#### 3.2.1. Cytotoxic Activity against MCF-7 Cell Line

MTT cell viability assay was used to evaluate the cytotoxic effects of furan-based compounds **2**–**7** on MCF-7 cancer cells and MCF-10a breast cancer cells. In a 96-well plate, cells were seeded at a density of 1 × 10^4^ at 37 °C for 24 h with 5% CO_2_. A total of seven different furan-based compounds **2**–**7** were tested and cultured with cells for 24 h before MTT solution at 5 mg/mL was administered and incubated for another four hours at 37 °C. Each well received 100 µL of dimethyl sulphoxide (DMSO) to dissolve the purple formazan. An ELISA plate reader (EXL 800, Waltham, MA, USA) measuring absorbance at 570 nm measured the formazan product’s color intensity, which depicted the cells’ growth state. At least three replicates were used to conduct the experiments, and the data are represented as the mean ± SD.

#### 3.2.2. Tubulin Inhibitions Assay

Compounds **4**, **7**, and podo were evaluated for their tubulin inhibitory activity according to manufacturer’s instructions [28].

#### 3.2.3. Cell Cycle Analysis of Test Compounds

At a density of 2 × 10^5^ cell per well, MCF-7 cells were collected and washed in PBS twice. At 37 °C and 5% CO_2_, the cells were then incubated. Compounds **4** and **7** were added to the medium for 48 h, washed twice in PBS, fixed with 70% ethanol, and rinsed with PBS. After that, for 15 min at 37 degrees Celsius, the medium was stained with DNA fluorochrome PI. Facs Calibur flow cytometer was used to evaluate the samples right away (Becton and Dickinson, Heidelberg, Germany).

#### 3.2.4. Annexin V FITC/PI Staining Assay

When treated for 48 h at their IC_50_ (µM), compounds **4** and **7** treated with MCF-7 cells at a density of 2 × 10^5^ per well were collected and stained with Annexin V-FITC/PI dye at 37 °C for 15 min in the dark. The FACS Calibur flow cytometer was immediately used to evaluate the samples (Becton and Dickinson, Heidelberg, Germany).

#### 3.2.5. ELISA Measurements of p53, Bax and Bcl2

Compounds **4** and **7** at their IC_50_ concentration (M) were shown to activate the p53, Bax, and Bcl-2 enzymes in MCF-7 cells using a previously published methodology [28].

## 4. Conclusions

In summary, a novel series of furan-based compounds was synthesized and its cytotoxic activity against the breast cancer cell line MCF-7 as well as the normal breast cell line MCF-10a was determined. Compounds **4** and **7** exhibited significant anticancer activity against a breast cancer cell line at a micromolar ratio, with IC_50_ values of 4.06 and 2.96 M, respectively. Additionally, cell cycle analysis demonstrated that compounds **4** and **7** induced cell cycle arrest at the G_2_/M phase and accumulated cells in the pre-G1 phase. The annexin-V/PI staining experiment suggests a role for apoptotic mechanisms. Simultaneously, compounds **4** and **7** significantly suppressed tubulin polymerization when compared with the reference compound. Additionally, compounds **4** and **7** had a strong pro-apoptotic activity via stimulation of the intrinsic apoptotic pathway. A considerable rise in the levels of p53 and Bcl-2, as well as a decrease in the levels of Bcl-2, when compared with control, corroborated this molecular pathway. As a result, furan-based derivatives may be considered as a scaffold for further structural optimization in the creation of anticancer drugs.

## Data Availability

Not applicable.

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
