# Peer review of "Design and Synthesis of Some New Furan-Based Derivatives and Evaluation of In Vitro Cytotoxic Activity"

_molecules, 2022, doi:10.3390/molecules27082606_

Round 1

Reviewer 1 Report

This manuscript reports the synthesis and cytotoxicity of furan derivatives. The work seems carefully done and the introduction, experiment design, results, and conclusions are clearly presented. The manuscript needs corrections as follows.

(1) Line 126: The SD (or SE) of the IC50 values of staurosporine against MCF-7 and MCF-10A and of compounds 4 and 7 against MCF-10A are very large. The authors should describe them correctly.

(2) Line 127: The authors should state that the values are expressed as means and SD (or SE) in the footnotes of Table 1 or in Experimental.

(3) Line 128: The data in Figure 3 is the same as the data in Table 1, so Figure 3 is not needed.

(4) Line 378: ‘100 ul’ should be changed to ‘100 uL’.

Author Response

Response to comments from Reviewer # 1:

This manuscript reports the synthesis and cytotoxicity of furan derivatives. The work seems carefully done and the introduction, experiment design, results, and conclusions are clearly presented. The manuscript needs corrections as follows.

We sincerely thank very much the reviewer for constructive criticisms and suggestions made to improve the manuscript, which were of great help in revising the manuscript. We have addressed all the editor and reviewer comments as below.

(1) Line 126: The SD (or SE) of the IC50 values of staurosporine against MCF-7 and MCF-10A and of compounds 4 and 7 against MCF-10A are very large. The authors should describe them correctly.

We apologize for the confusion, the mistake was corrected and highlighted.

(2) Line 127: The authors should state that the values are expressed as means and SD (or SE) in the footnotes of Table 1 or in Experimental.

Done as suggested by the reviewer, and it is highlighted.

(3) Line 128: The data in Figure 3 is the same as the data in Table 1, so Figure 3 is not needed.

Done as suggested by the reviewer.

(4) Line 378: ‘100 ul’ should be changed to ‘100 uL’.

We apologize for the mistake; the mistake was corrected and highlighted

Reviewer 2 Report

The article describes synthesis and mechanisms of cytotoxic activity of furan-based derivatives. In general, it is well prepared, however I have some remarks.

-to underline the low toxicity of compounds 4 and 7, one should calculate SI (selectivity index) and add it to the text

-the Figure 3 gives the same data as Table 1 and is unnecessary

-Figure 4 – there is no reference compound in it- what is the level of tubulin inhibition presented by STU? The same for other cytotoxicity tests – the results for tested compounds were not compared with STU action (cell cycle arrest, apoptosis).

-there is no coma after (or before) the word that

Author Response

Response to comments from Reviewer # 2:

The article describes synthesis and mechanisms of cytotoxic activity of furan-based derivatives. In general, it is well prepared, however I have some remarks.

We sincerely thank very much the reviewer for constructive criticisms and suggestions made to improve the manuscript, which were of great help in revising the manuscript. We have addressed all the editor and reviewer comments as below.

-to underline the low toxicity of compounds 4 and 7, one should calculate SI (selectivity index) and add it to the text

The reviewer comment was highly appreciated. The SI of compounds 4 and 7 was calculated and a paragraph was added to the text and it is highlighted.

-the Figure 3 gives the same data as Table 1 and is unnecessary

As suggested by the reviewer, Figure 3 is deleted.

-Figure 4 – there is no reference compound in it- what is the level of tubulin inhibition presented by STU? The same for other cytotoxicity tests – the results for tested compounds were not compared with STU action (cell cycle arrest, apoptosis).

The reviewer comment was highly appreciated. This study was a preliminary discriminatory study conducted in order to estimate the cytotoxicity of the prepared compound and their percentage of tubulin inhibition and the full biological study with the compounds will be conducted in further publications

-there is no coma after (or before) the word that

We apologize for the mistake, all the manuscript was revised and coma after (or before) the word that is removed.

Reviewer 3 Report

The manuscript submitted for review is suitable for publication in Molecules. Only minor editorial corrections required. The introduction should be corrected and it should be further explained why these tumor cell lines have been used. The diagrams were repeated in the work, eg delete one of the diagrams in figure 3. Delete figure 2 completely. 

Author Response

Response to comments from Reviewer # 3:

The manuscript submitted for review is suitable for publication in Molecules. Only minor editorial corrections required.

We sincerely thank the reviewer for constructive criticisms and comments, which were of great help in revising the manuscript. We really appreciate the suggestions made to improve the manuscript.

The introduction should be corrected and it should be further explained why these tumor cell lines have been used.

The reviewer comment was highly appreciated. The authors utilized the MCF-7 cell line as the study was based on a preliminary study and the main focus of the study was to prepare compounds of improved cytotoxicity against MCF-7 cells and its mentioned in the manuscript and highlighted.

References:

  1. https://doi.org/1016/j.ejmech.2018.04.012
  2. https://doi.org/10.1134/S1070363219030204
  3. https://doi.org/3390/ph14101021

The diagrams were repeated in the work, eg delete one of the diagrams in figure 3. Delete figure 2 completely.

As suggested by the reviewer. Figure 3 is deleted, but Figure 2 represents the design of the present work, I thank it will make the manuscript easy for readers.

Round 2

Reviewer 2 Report

The article is suitable for publication.